# Effectiveness of Snail Slime in the Green Synthesis of Silver Nanoparticles

**DOI:** 10.3390/nano12193447

**Published:** 2022-10-01

**Authors:** Maria Francesca Di Filippo, Valentina Di Matteo, Luisa Stella Dolci, Beatrice Albertini, Barbara Ballarin, Maria Cristina Cassani, Nadia Passerini, Giovanna Angela Gentilomi, Francesca Bonvicini, Silvia Panzavolta

**Affiliations:** 1Department of Chemistry “G. Ciamician”, University of Bologna, Via Selmi 2, 40126 Bologna, Italy; 2Department of Industrial Chemistry “Toso Montanari”, University of Bologna, Via Risorgimento 4, 40136 Bologna, Italy; 3Department of Pharmacy and BioTechnology, University of Bologna, Via S. Donato 19/2, 40127 Bologna, Italy; 4Center for Industrial Research-Advanced Applications in Mechanical Engineering and Materials Technology CIRI MAM, University of Bologna, Viale del Risorgimento 2, 40136 Bologna, Italy; 5Center for Industrial Research-Fonti Rinnovabili, Ambiente, Mare e Energia CIRI FRAME, University of Bologna, Viale del Risorgimento 2, 40136 Bologna, Italy; 6Health Sciences and Technologies—Interdepartmental Center for Industrial Research (HST- ICIR) Alma Mater Studiorum—University of Bologna, 40064 Ozzano dell’Emilia, Italy; 7Department of Pharmacy and Biotechnology, University of Bologna, Via Massarenti 9, 40138 Bologna, Italy

**Keywords:** silver nanoparticles, green synthesis, snail slime, antibacterial activity, biogenic pathway

## Abstract

The development of green, low cost and sustainable synthetic routes to produce metal nanoparticles is of outmost importance, as these materials fulfill large scale applications in a number of different areas. Herein, snail slime extracted from *Helix Aspersa* snails was successfully employed both as bio-reducing agent of silver nitrate and as bio-stabilizer of the obtained nanoparticles. Several trials were carried out by varying temperature, the volume of snail slime and the silver nitrate concentration to find the best biogenic pathway to produce silver nanoparticles. The best results were obtained when the synthesis was performed at room temperature and neutral pH. UV–Visible Spectroscopy, SEM-TEM and FTIR were used for a detailed characterization of the nanoparticles. The obtained nanoparticles are spherical, with mean diameters measured from TEM images ranging from 15 to 30 nm and stable over time. The role of proteins and glycoproteins in the biogenic production of silver nanoparticles was elucidated. Infrared spectra clearly showed the presence of proteins all around the silver core. The macromolecular shell is also responsible of the effectiveness of the synthesized AgNPs to inhibit Gram positive and Gram negative bacterial growth.

## 1. Introduction

In recent years, research has been increasingly interested in the development of nanomaterials with tailored properties for applications in the biomedical and pharmaceutical fields, for drug delivery and biosensing. In particular, some innovative therapies in the field of wound management started using nanomaterials, which are slightly invasive, can be passive or targeted, and should result in minimal side effects. Moreover, various methods for the synthesis and fabrication of different kind of nanocomposites have been exploited [1,2].

Among the nanomaterials available for the healing of infected wounds (i.e., inorganic nanomaterials, organic and hybrid nanomaterials, and nanofibers), silver nanoparticles (AgNPs) have been found to have exceptional antimicrobial, and antiviral properties [3,4] compared to silver ions solutions. AgNPs show also prominent inhibitory activity against a broad spectrum of resistant strains in Gram-positive and Gram-negative bacteria [5,6,7,8,9,10]. Contrary to bactericide effects of ionic silver, the antimicrobial activity of colloid silver particles is influenced by particles dimensions: the smaller the particles, the greater the antimicrobial effectiveness [11]. Therefore, in developing routes of synthesis, an emphasis was made to control the size of silver nanoparticles.

Chemical methods, in which reduction or co-precipitation processes take place for the generation of nanoparticles, have been investigated since ancient times [12]. However, the main drawbacks of these synthetic routes are the use of harsh chemical conditions and organic solvents, and the production of toxic by-products during the synthesis and/or functionalization of nanoparticles. For this purpose, many studies are focused on the research of biocompatible procedures and methods, supporting the use of natural extracts for the rapid and green synthesis of nanoparticles [13]. Greener methods employing a variety of biological agents (plant extracts, bacterial, and fungal derived compounds) have been used to convert silver salts to nanoparticles [14,15,16].

The role of some peptides and proteins in the reduction of silver ions has been reported in literature [17,18,19], even if in most cases the coupling of proteins with a reducing agent, like ascorbic acid or sodium borohydride, is mandatory [20,21]. Some papers also evidenced the effect of selected amino acids, namely tyrosine, glutamic acid and aspartic acid in the growth of anisotropic AgNPs. For example, Xie and colleagues provided evidence that peptide sequences in algal proteins directly influence the formation and morphologies of Ag nanoparticles [15].

Due to the growing interest and the wide field of applications of AgNPs, a simple, sustainable, effective and low-cost route for the synthesis of silver nanoparticles in accordance with the green chemistry philosophy has been successfully achieved in our work thanks to the use of snail slime as reducing and stabilizing naturally derived agent.

Snail slime is a mixture of active compounds such as proteins, glycoproteins, glycosaminoglycans (GAGs), fatty acids, polyphenols, vitamins, glycolic acid, allantoin, minerals [22,23,24] and carbohydrates [25]. The peculiar composition of snail slime has stimulated several studies, putting into evidence its beneficial effects in different pathologies [22,26]. Proteins contained in snail slime have been successfully used for the reduction of HAuCl_4_ to gold nanoparticles, exhibiting wound healing and anti-inflammatory properties [14,26]. Recently, composites AgNPs have been prepared mixing silver nitrate with diluted snail mucus and using ascorbic acid as reducing agent and polyvinyl pyrrolidone as stabilizer and their antimicrobial and anticancer activities have been assessed [26].

The green procedure we propose for the synthesis of AgNPs employs only snail slime, obtained from *Helix Aspersa* snails: thanks to the peculiar composition of snail slime, we demonstrated its effectiveness to act both as reducing and stabilizing agents for the effective synthesis of AgNPs, avoiding the use of additional and toxic agents, and employing aqueous solutions at room temperature. Furthermore, the known antibacterial properties of AgNPs are strongly enhanced by the presence of slime macromolecules wrapping the nanoparticles.

The influence of different preparation conditions, such as silver nitrate concentration, volume of the reagents and process temperature, were investigated. Furthermore, prior to the AgNPs synthesis, snail slime has been dialyzed, characterized by UV-Vis and IR spectroscopy, and analyzed in terms of dry residue and protein content. Then, the collected AgNPs were characterized both in solution and in dry state by means of microscopy and spectroscopic techniques, and their stability over time and the enhancement of their antibacterial activity were proved.

## 2. Experimental Methods

Gelatin type B, sodium chloride (>99.5%, MW = 58.44 g/mol), bicinchoninic acid, ethanol (>99.8%), CuSO_4_∙5H_2_O salt, and KH_2_PO_4_ salt were supplied by Sigma Aldrich. D-(+)-Glucose anhydrous (>99.5%, MW = 180,16 g/mol) was purchased by Fluka Biochemika.

### 2.1. Collection and Purification of Snail Slime

The snail slime obtained from *Helix Aspersa Muller* snails was kindly donated by ‘lumacamadonita’ (Palermo, Italy). Commercial solution, obtained by dilution of extracted slime with purified water, was used.

Snails are farmed outdoors, on land, and fed with fresh vegetables. Snails are collected three times per year, washed with water and placed inside the patented extractor machine EXTRACTA. After snails disinfection by ozone treatment, the slime is extracted by means of mechanical stimulation, a cruelty-free method, that allows to obtain the product with a pH between 5 and 7 (www.lumacamadonita.it, accessed on 6 September 2022).

Prior to use, the slime was subjected to dialysis by means of a dialysis tubing cellulose membrane (Merk) with a cut-off of 12–14 kDa, to remove the additives added by the retailer. The membranes were previously washed with abundant distilled water to remove primary glycol used as a humectant. A known volume of liquid snail slime was poured inside the conditioned membrane, maintained 24 h in distilled water under stirring; the water was replaced twice.

The dialyzed samples were stored at −19°C until use, while some samples were lyophilized after freezing and stored between 0 and 4 °C.

The results of the characterizations performed on snail slime as received and after dialysis are reported in Table 1.

UV-vis spectra were collected both on as received slime properly diluted and on the freeze dried and re-suspended dialyzed material, by using a Cary 60 UV-Vis spectrophotometer. The measurement of the protein content was performed by bicinchoninic acid assay, following a procedure previously reported [27]. Infrared spectra were collected on KBr pellets containing the dried sample, as reported below.

### 2.2. Synthesis of AgNPs

To obtain AgNPs using dialyzed snail slime, several trials were performed to find the best synthetic conditions: the effects of the starting AgNO_3_ concentration, the amount of snail slime and the temperature conditions were evaluated as reported in Table 2. Variable volumes of ultrapure water were added to the reagents to maintain the final volume equal to 5 mL. The synthesis was performed in closed vials and left under stirring for 5 days, during which the color of the solution turned from transparent to deep yellow, with the concomitant appearance of the plasmonic band at around 400 nm daily monitored by UV-Vis during the whole period. AgNPs-containing solutions were then stored in dark to avoid any photochemical reaction.

For the synthesis of AgNPs used as control (AgNPs-Ref), a protocol found in literature was used [19]. Briefly, 0.16 g of gelatin type B were solubilized into 15 mL of ultrapure H_2_O and 0.8 mL of 1.0 M AgNO_3_ solution were added to the dissolved gelatin under continuous stirring. Then, 1.6 mL of glucose solution (2.0 M) were added to the Ag^+^/gelatin solution. The mixture was kept under stirring for 48 h at the selected temperature and then stored in the dark.

### 2.3. Characterization of AgNPs

Visible absorption spectra of the solutions containing AgNPs were recorded using a Varian CARY 5 UV-Vis spectrophotometer in the range 300–800 nm. The measurements were performed using a cuvette with 1 cm path length.

Zeta potential was measured by Dynamic Light Scattering (DLS) using a Malvern Zetasizer Nano ZS instrument with laser at 532 nm and divergent degree of 173°. Each analysis was performed in triplicate.

The obtained AgNPs were observed by using a Philips CM 100 Transmission Electron Microscope operating at 80 kV (TEM) and a ZEISS Leo 1530 Scanning Electron Microscope (SEM). The nanoparticles were deposited by drop casting onto Cu-grids covered by a Formvar layer (400 mesh, Cu—TED PELLA, INC.) or onto Al-stubs and analyzed after solvent evaporation at room temperature. The size distribution of the AgNPs was estimated by TEM images using Digimizer Image Analysis Software. At least 100 nanoparticles were measured for each composition.

To estimate the amount of silver involved in the formation of the NPs, the quantification of unreacted Ag^+^ ions after AgNPs synthesis was carried out by means of conductometric titration (XS COND 51 Conductivity Meter). AgNPs-containing colloidal suspensions were centrifuged by using Millipore Amicon Ultra-15 50K Centrifugal Devices (Awel centrifugation MF 20-R; 10,000 rpm at room temperature for 15 min) to separate the nanoparticles from the solution, which was then titrated using NaCl (10^−3^ M). The titrant was added to the centrifuged solution under stirring, and the volumes of NaCl added vs. conductivity values (μS/cm) were plotted.

Fourier-transform infrared spectroscopy (FTIR, Nicolet IS10 spectrophotometer) was performed with a spectral resolution of 4 cm^−1^ and 70 scans from 4000 to 500 cm^−1^ on samples prepared as KBr disks. Freeze-dried snail slime samples before and after dialysis were used. AgNPs separated by ultracentrifugation were washed twice and freeze-dried before collecting infrared spectra. Omnic software (Thermo Electron Corp., Woburn, MA, USA) was used for data processing and baseline correction.

### 2.4. Assessment of Antibacterial Activity

*Staphylococcus aureus* (ATCC 25923) and *Escherichia coli* (ATCC 25922) were selected as representative Gram-positive and Gram-negative bacterial models, respectively, to test the antibacterial properties of the AgNPs samples. A well-established broth microdilution method was carried out according to the guidelines established by a number of International committees for Antimicrobial Susceptibility Testing [28,29]. Briefly, the bacterial inocula were prepared in PBS (phosphate buffer saline) and adjusted to an Optical Density at 630 nm (OD 630 nm) of 0.08–0.1. The suspensions were diluted 1:200 in Mueller-Hinton (MH) broth and then incubated with serial 2-fold dilutions of samples, starting from a nominal concentration of 460 µM of Ag^+^. Several wells were reserved in each microplate for negative and positive controls, including the dialyzed snail slime, the glucose gelatin/solution and AgNO_3_, as source of free Ag^+^ ions. The microplate containing a final volume of 200 μL/well was incubated at 37 °C for 24 h, and subsequently, the OD 630 nm was measured using a microplate reader spectrophotometer. Growth percentage values of the tested bacteria at the different experimental conditions were determined as relative to the untreated positive controls (bacterial cells incubated in MH broth). Fifty-percent inhibitory concentration (IC_50_) values were determined as the inflection points of the four-parameter dose-response curves obtained by plotting the percentage values of the bacterial growths vs. the logarithm of the Ag-sample concentrations. The one-way analysis of variance (ANOVA) followed by Tukey’s Multiple Comparison Test was used to compare IC_50_ values of the tested samples; statistically significant differences were determined at *p* < 0.05.

### 2.5. Cell Viability and Proliferation

Vero cell line (ATCC CCL-81) was used to investigate the overall effect of the samples on mammalian metabolism, as these epithelial non-malignant cells are a well-defined experimental setting for cytotoxicity studies [30,31,32]. Briefly, cells were grown in RPMI-1640 medium supplied with 10% FBS (Fetal Bovine Serum), 100 U/mL penicillin, and 100 µg/mL streptomycin), at 37 °C and 5% CO_2_. For experiments, cells were seeded into 96-well plates at 10^4^ cells/100 µL, then incubated at 37 °C for 24 h. Following washes with PBS, cell monolayer was incubated with 100 µL of medium containing the samples at different concentrations, ranging from 460 µM to 3.6 µM of Ag^+^.

The cell viability was assessed by a WST8-based assay according to the manufacturer’s instructions (CCK-8, Cell Counting Kit-8, Dojindo Molecular Technologies, Rockville, MD, USA). After 48 h of incubation, culture medium was removed from each well, the monolayer was washed with PBS, and 100 µL of fresh medium containing 10 µL of CCK-8 solution were added. Following a 2 h at 37 °C, the OD 450 nm was measured and data were calculated as percentages relative to the untreated positive controls (Vero cells incubated in regular medium). Fifty-percent inhibitory concentration (CC_50_) values were determined as previously described for IC_50_ values.

## 3. Results and Discussion

Commercial snail slime are solutions, diluted with water, of the pure extract obtained by different extractive techniques from living snails. Snails farming is a growing agricultural activity all over the world, and a representative expression of circular economy. In fact, after several cycles of slime extraction, snails are sold as food and their shells employed as a source of biological calcium carbonate. Several methods for slime extraction, that mainly use acidic or neutral stimulating solutions, are proposed and patented. It is known that the extraction method can affect the final composition of the snail, as well as the snail breeding and feeding [33]. The active substances present in the mucus make it a unique natural product, impossible to reproduce in the laboratory with synthetic chemical compounds.

In our work, we used a commercial snail slime extracted without the use of stimulating substances. As the extraction method can influence the slime composition, before using we performed a characterization of the slime, focused in particular to the quantification of the proteic macromolecules.

Prior to use, the commercial snail slime solution was dialyzed and characterized to evaluate its dry residue and protein content. Dialysis was carried out to remove chloride ions, and additives and/or preservatives added by the producers after extraction: furthermore, the cut-off employed (12–14 kDa) allowed to enrich the slime by proteins with a higher molecular weight, in tune with the findings reported by Xie et al., that demonstrated the ineffective action of proteins less than 7 kDa in the synthesis of AgNPs [15]. The pH of the dialyzed solution was 7.03.

### 3.1. Snail Slime Characterization

The UV-Vis spectra collected on snail slime before and after dialysis are reported in Figure 1. In the absorption spectrum of the as-received sample, the first band, centered at 207 nm, is due to the peptidic backbone, while the second one, at higher wavelengths, arises from the side groups of the amino acid residues. In particular, cysteine, with a maximum of absorption at 250 nm, and aromatic amino acids like tyrosine, tryptophan and phenylalanine are responsible of this absorption band, centered at 256 nm for the as-received slime. The red shift at 266 nm observed in the dialyzed slime could account for a decrease of Cys residues and an increase of proteins containing aromatic residues, as their absorption maximum ranges between 250 and 280 nm. As the thiol group is able to complex silver ions, thus preventing their reduction, a decrease of Cys residues can favorably promote AgNPs formation. Furthermore, the presence of high tyrosine content in the composition of snail slime should be very important, because Xie et al. [15] clearly demonstrated its role in the reduction of silver ions.

The dry residue was evaluated both on as received slime and on the dialyzed one, by weighting the sample before and after lyophilization using the following formula:$$dry\;residue(\%)=\frac{W_{f}}{W_{i}}\times 100$$where *W_f_* and *W_i_* are the final weight of the lyophilized sample and that of the liquid snail slime before lyophilization, respectively. The values, obtained as an average of at least 5 samples, are reported in Table 1.

The dry residue of the as received snail slime accounts for 1.1%, and it is in the range of the values reported in literature, that fall between 0.2 and 3.0% [22,33]. After dialysis, the weight of the recovered sample is significantly lower: indeed, during this step all the compounds with a molecular weight lower than 14 kDa pass through the membrane and are lost.

The protein content was evaluated by the bicinchoninic acid assay (BCA) that is a method highly sensitive and tolerant to interfering species method, allowing quantification of both proteins and glycoproteins [34]. The BCA assay was performed on liquid and on freeze-dried samples both before and after the dialysis, and the results are reported in Table 1.

The amount of total proteins and glycosylated proteins in the snail slime as received accounts for 1.80 mg/mL, a value falling in the interval reported in other studies for *Helix Aspersa* snails [22,33]. However, it is well known that materials of natural origin are characterized by a huge variability, and a variable protein content could be also attributed to the different extraction methods other than to the farming and breeding of the snails. The concentration of proteins in the freeze-dried sample after dialysis is higher than that in the as received sample, since the dialysis process eliminates all the compounds below the membrane threshold: in particular, the cut-off employed allowed us to obtain a dialyzed sample containing nearly 100% of protein macromolecules, well evidenced also by infrared spectra showed in Figure 2.

The broad band centered at 3400 cm^−1^ is attributed to -OH stretching, while the double band centered at 2950 cm^−1^ is attributed to the -CH_2_- stretching. The infrared spectrum of snail slime as received shows several broad bands between 1453–1500 cm^−1^ which are no longer detectable in the dialyzed sample thus representing further evidence of the presence of additives at very high concentration, as supposed by comparing the weight of the dry residue to the protein content. The presence of protein macromolecules in IR spectra can be undoubtedly revealed by the appearance of two absorption bands at 1653 and 1545 cm^−1^ attributed to Amide I and Amide II, respectively; the absorption associated with the Amide I originates from the stretching vibrations of the C=O bond of the amide, while absorption associated with the Amide II is mainly due to the bending vibrations of the NH bond. Both these bands become very strong after dialysis, clearly showing the increase of protein content in the residue. The amide III band at 1238 cm^−1^ is also diagnostic of the presence of proteins since originates from C-N stretching [35]. The bands between 1415 and 1380 cm^−1^ are due to the carboxylate moieties associated with the acid derivatives of sugars and amino acid chains [14], while the broad band centered at 1100 cm^−1^ can be assigned to C-O bonds, thus confirming the presence of glycosylated proteins.

The dialyzed snail slime, richest in proteins, was then used for the synthesis of AgNPs.

### 3.2. Synthesis and Characterization of Silver Nanoparticles

Reference AgNPs (AgNPs-Ref) were prepared by a green synthesis reported in literature [19], involving heating at 60 ºC for 2 days and the use of glucose and gelatin as reducing and stabilizing agents, respectively. The characterization of AgNPs-Ref was performed as comparison and some results are discussed in the following.

In order to find the best experimental conditions, the synthesis of AgNPs using snail slime was carried out both at room temperature (25 °C) and at 60 °C, by varying the silver nitrate concentration and its relative amount with respect to snail slime, as summarized in Table 2.

With the aim to set up an effective, simple, biogenic, and sustainable synthetic route, the first attempts were made at room temperature by mixing 1.5 mL of AgNO_3_ 10^−3^ M with different volumes of dialyzed snail slime. A color change, from colorless to pale yellow, began to appear for samples C and A after 2 days from mixing, becoming more intense over time. On the contrary, solution E maintained a pale purple color up to five days after mixing. UV-Vis spectroscopy has been used just as a qualitative method to assess the formation of AgNPs. According to the literature, the UV-Vis spectrum of silver nanoparticles, known as the surface plasmon absorption band, gives a strong band, usually centered at wavelengths between 395 and 445 nm: the maximum can vary as a consequence of the particle’s size [8,36].

From the UV-Vis spectra collected on samples A, C and E and reported in Appendix A, a strong band centered at 428 nm, indicative of the presence of AgNPs, was obtained only for synthesis A, where the volume ratio Ag/snail was 1.5/1. A less pronounced band was obtained for sample C (Ag/snail volume ratio = 1.5/2), while sample E (Ag/snail volume ratio = 1.5/0.5) gave a broad and poorly intense band.

Since several syntheses of AgNPs reported in literature [18,19,21] evidenced the favorable impact of the temperature, some trials were conducted by raising up the temperature to 60 °C (samples B, D and F). However, a higher temperature did not improve the AgNPs formation: as reported in Appendix A only for trial B a very broad band composed of two poorly resolved peaks was obtained.

Driven by the good results obtained at room temperature, AgNPs synthesis was performed by using two Ag/snail volume ratios just tested (1.5/1 and 1.5/2) but increasing the silver nitrate concentration. The experiments G and H also produced AgNPs and the onset of the reaction, revealed by the color turning to yellow, started soon after the mixing of the reactants, even if the NPs concentration increases over time, as evidenced by spectra collected daily (see for example the UV-Vis spectra collected from sample H after 3 and 5 days from mixing, Appendix A).

UV-Vis spectra acquired on samples G and H after 5 days from mixing were superimposed to that obtained from sample A and reported in Figure 3 together with the optical pictures of the syntheses. The plasmonic bands of the samples A, H, G largely overlap, with maxima around 440 nm for H and G.

To evaluate the yield of the reaction, a conductometric titration has been performed adapting a method reported in literature [37] to measure the amount of free silver ions in solution after NPs formation. Synthesized AgNPs were centrifuged by means of Amicon tubes, equipped with 50 kDa membrane. The solution passed through the membrane was titrated by properly diluted NaCl and the titration followed by a conductivity meter.

The reaction yield estimated that about 50% of silver ions were reduced in the synthesis A and H, while a value of 65% was obtained for sample G. As comparison, we evaluated that the yield of AgNPs-Ref, obtained using glucose as reducing agent and a temperature of 60 °C, was around 68%, while in literature Gorup et al. [37] reported a 50% yield from the synthesis of AgNPs by using a 10^−3^ M silver solution and citric acid at 90 °C. These data highlighted the excellent performance of snail slime even at room temperature.

The biosynthesized samples A, G and H were also analyzed by DLS and presented an average hydrodynamic diameter value of 156 nm with a polydispersity (PDI) below 0.3 in all cases, indicating the absence of larger agglomerates. An average zeta potential of −19.2 mV proved the stability and the relative homogeneity of the synthesized nanoparticles [38]. These data are comparable to those found for snail slime alone (201 nm particle size, −28.5 mV zeta potential).

A morphological characterization of the obtained NPs has been performed by means of electron scanning microscopy (SEM) and transmission microscopy (TEM). Images were collected soon after the synthesis and are reported in Figure 4 together with particle’s size distribution obtained from TEM images.

Transmission microscopy revealed a strong influence of the synthesized condition on the size of the nanoparticles. The formation of regularly round-shaped AgNPs with a minimum and maximum size ranging from 5 to 80 nm was observed in sample A, while sample G and H appeared smaller and less polydisperse. In particular, AgNPs obtained by method H showed a mean size of 22 ± 9. These results underlined the influence of the AgNO_3_/slime volume ratio on the synthesis. In fact, the NPs nucleation, growth and stabilization are the result of several factors, including ionic force and density of nuclei at the beginning of the process. A balanced condition between all these contributions decreases the concentration of silver ions and of new nuclei, thus enhancing NPs growth. Using TEM, only the size of the NPs metallic core can be estimated; while analyzing the AgNPs in the colloidal solution the combination of the Ag core with the hydrated protein layer are detectable [18,36]. The snail slime wrapping the core has a dynamic character and fluctuates in the bulk medium, thus all the NPs display a similar hydrodynamic diameter, as evidenced by DLS measurements.

At the scanning microscope, AgNPs aggregates made of round-shaped distinct nanoparticles are clearly visible. Reference AgNPs were also characterized, and images are reported in Appendix A.

To evaluate the stability of AgNPs over time, UV-Vis spectra were collected from colloidal suspensions of samples A, G and H at regular intervals after the preparation. The superimposition of these spectra with those collected after 5 days from the beginning of the synthesis did not reveal any shift of the maximum (see Appendix A, where spectra collected from synthesis H after five and fifty days are superimposed) thus supporting the persistent role of snail slime in nanoparticles stabilization.

Separation of AgNPs from the starting solution could be useful to widen their field of application: however, aggregation could occur during this stage, thus suppressing the peculiar properties of nanomaterials. To assess the recovery of a fine AgNPs dispersion, silver nanoparticles obtained by synthesis H were centrifuged by means of 50 kDa membrane-equipped testing tubes and re-suspended in double distilled water. The UV-Vis spectrum collected on resuspended AgNPs still shows a maximum peak around 440 nm revealing that no aggregation occurs during the separation stage.

AgNPs (synthesis H) separated by centrifugation were characterized by means of IR spectroscopy. In Figure 5 the superimposition of infrared spectra, collected from AgNPs and from dialyzed snail slime, well evidenced the presence of an organic matrix, surrounding the silver core, with some interesting hints.

The characteristics bands belonging to proteins (Amide I and II), well separated in the infrared spectrum of dialyzed slime appeared fused together, and more bands appeared in the range 1400–1300 cm^−1^ indicating the electrostatic interaction between AgNPs and functional polar groups (OH, N–H, and C=O) of the macromolecules [21]. Most interestingly, a small but detectable band appeared at 1724 cm^−1^: according to the literature [15], this band was attributed to carbonyl stretching, arising from the formation of a phenoxide structure due to the oxidation of the tyrosine phenolic group. As previously stated, tyrosine was demonstrated to be able of reducing silver: the presence of Tyr in the dialyzed sample, just hypothesized from the analysis of UV-Vis spectra (see Figure 1) was thus confirmed by this further evidence.

These results evidenced that snail slime alone is able to promote the formation and stabilization of silver nanoparticles, without the the need of additional reducing or stabilizing agents as previously reported in literature [26]. These findings further support the employment of snail slime in green and sustainable processes.

### 3.3. Antibacterial Assay

The effectiveness of the newly devised AgNPs was assessed in vitro against two bacterial strains and evaluated in comparison with AgNPs-Ref. For this purpose, *S. aureus* and *E. coli* were incubated for 24 h with serial 2-fold dilutions of samples G, H and AgNPs-Ref starting from a nominal concentration of 460 µM of Ag^+^ while the highest tested concentration for sample A was 46 µM. As controls, bacterial strains were also assayed with the corresponding amount of dialyzed snail slime and glucose/gelatin reactants without AgNO_3_ solution as well as with serial 2-fold dilutions of AgNO_3._

Growth percentage values were spectrophotometrically evaluated, and data were expressed relative to the untreated controls, as neither the dialyzed snail slime nor the glucose/gelatin solution proved to reduce bacterial proliferation.

The mean IC_50_ values ranged from 266.8 µM to 390.6 µM for *S. aureus* and from 10.9 µM to 15.1 µM for *E. coli* (see Figure 6). To note that the IC_50_ value of sample A was not determined for *S. aureus* because the AgNPs were prepared using 300 µM of AgNO_3_, tested starting from 46 µM, and the Gram-positive strain are greatly resistant to AgNPs compared to *E. coli*. This finding has been previously reported for other Ag-based compounds [39,40] and it could be ascribed to a different interaction of the NPs with the bacterial cell wall of Gram positive and Gram-negative bacteria.

A comprehensive evaluation on the antibacterial activity of the newly synthetized AgNPs was therefore carried out for *E. coli*. The obtained results confirm that the herein developed methodology is a useful approach to produce bioactive NPs as samples A, H, and G strongly inhibited *E. coli* growth and at a similar extent of the reference AgNPs sample. In addition, samples H and G, containing the same nominal concentration of Ag^+^, displayed a slightly different antibacterial potency (IC_50_ 15.1 ± 6.1 µM and 10.9 ± 3.5 µM, respectively, see Appendix A) possibly suggesting that NPs produced in presence of a higher volume of snail slime could determine an improved interaction of the AgNPs with bacterial structures and/or an increased penetration of the released Ag^+^ into the bacterial cell. As coating, size, morphology, and polydispersity of NPs are all parameters impacting nanoparticle antimicrobial performance, it is possible to speculate that sample H has been produced by using the optimal green synthetic procedure enabling strong antibacterial activity.

To better characterize the biological properties of the AgNPs, sample G, the dialyzed snail slime, and AgNO_3_, as source of free Ag^+^ ions, were tested on Vero cells. Indeed, the cytotoxicity of Ag^+^ and of several Ag-derived materials is a well-known feature, commonly exploited when these products are used as anticancer drugs [41,42]. For this reason, the safety profile on normal cells was included in the present study with the aim of ascertaining the therapeutic window of the bioactive AgNPs as antimicrobial agents. Remarkably, the dialyzed snail slime didn’t interfere with cell metabolism at all tested dilutions suggesting its in vitro biocompatibility, while both samples G and Ag+ affected the viability of Vero cells with CC_50_ values of 68.1 ± 0.6 µM and 24.9 ± 0.1 µM, respectively. The selectivity index (SI = CC_50_/IC_50_) was therefore determined for sample G (SI = 4.5) and Ag+ (SI = 1.7) indicating the improved selectivity of the inhibitory activity of the devised AgNPs towards *E. coli*.

## 4. Conclusions

The use of purified snail slime, containing almost exclusively proteins and glycoproteins, allowed the formation of biogenic silver nanoparticles without using other reagents and under very mild conditions of pH and temperature: in fact, the best results were obtained at room temperature and at neutral pH. Tyrosine residues belonging to the protein chain have been proved to drive the reduction of silver ions, while the presence of macromolecules all around the nanoparticles played a fundamental role in their stabilization over time, thus preventing aggregation up to fifty days. Even after separation from the starting solution and redispersion into aqueous medium no significative changes in the silver nanoparticles are evident.

The herein produced AgNPs proved to be effective in inhibiting bacterial growths as well as the AgNPs used as reference sample. The use of snail slime in the synthesis represents a successful application of green procedures and circular economy principles and has an impact on the potency of these antibacterial agents. All things together, we can state that snail slime, obtained from *Helix Aspersa* snails, plays a crucial role as reducing and stabilizing agents in the synthesis of bioactive AgNPs.

## Figures and Tables

**Figure 1 nanomaterials-12-03447-f001:**
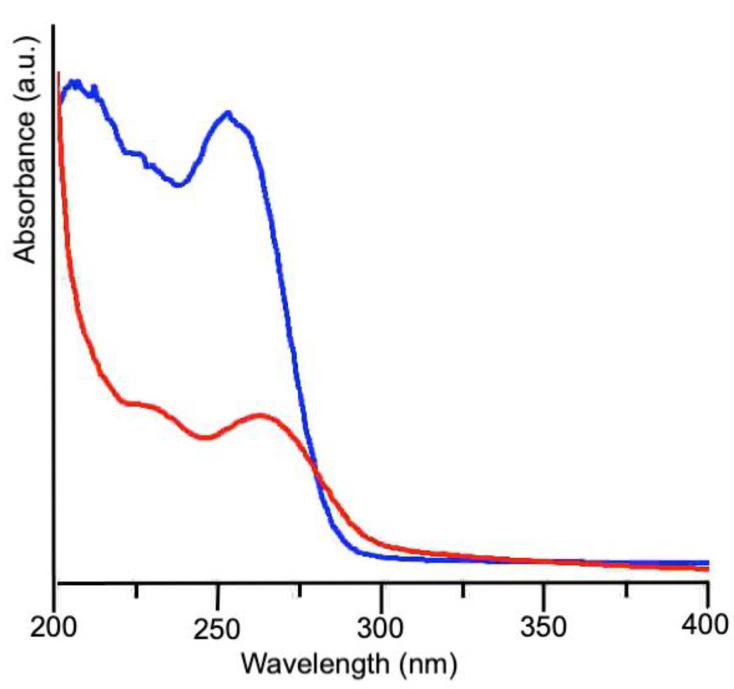
UV-vis spectra of snail slime as received (blue line) and after dialysis (red line).

**Figure 2 nanomaterials-12-03447-f002:**
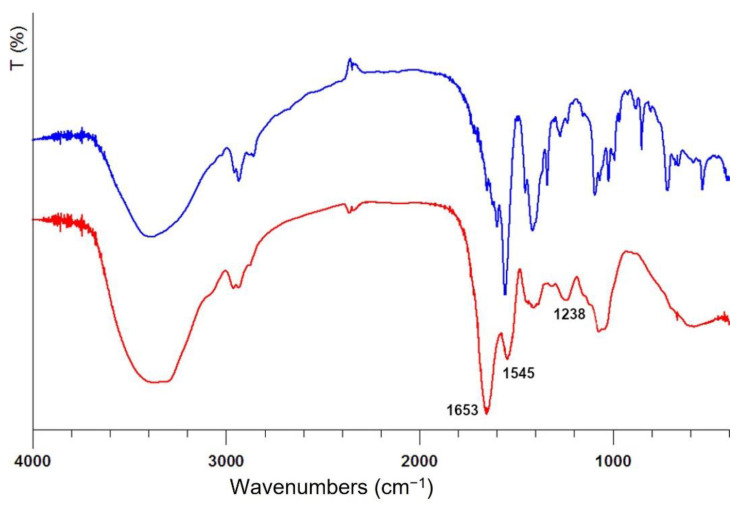
Infrared spectra of freeze-dried snail slime solution: as received (blue line) and after dialysis (red line). The wavenumbers of the most prominent bands belonging to proteins are reported.

**Figure 3 nanomaterials-12-03447-f003:**
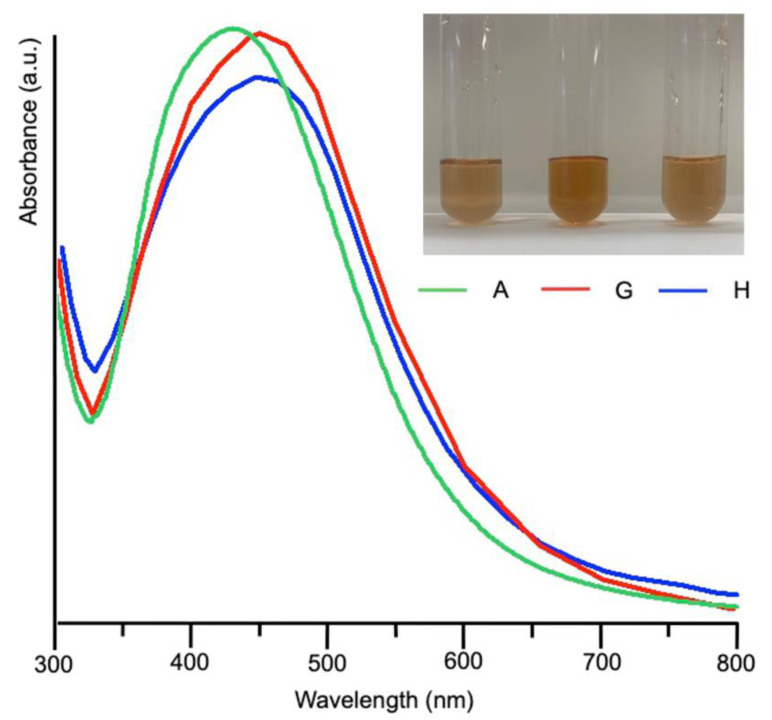
UV-Visible spectra obtained from samples A, G and H after five days from mixing and their relative pictures.

**Figure 4 nanomaterials-12-03447-f004:**
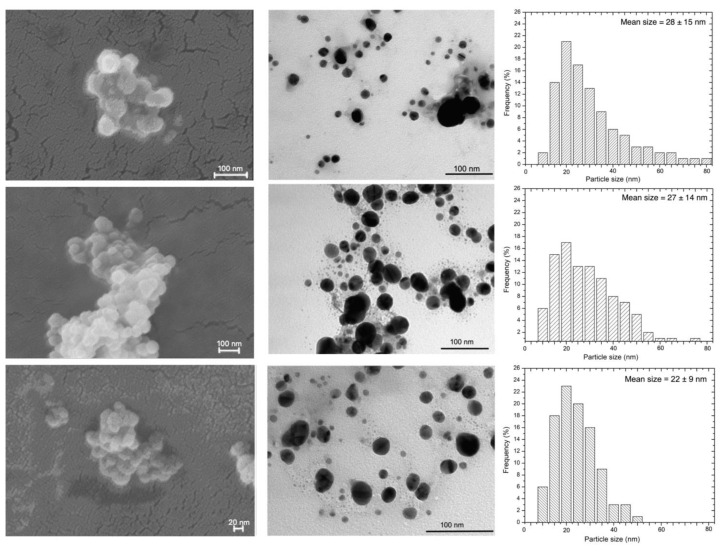
SEM micrographs (**left** column), TEM micrographs (**central** column) and AgNPs size distribution determined by TEM images (**right** column) for the samples A, G and H (from top to bottom).

**Figure 5 nanomaterials-12-03447-f005:**
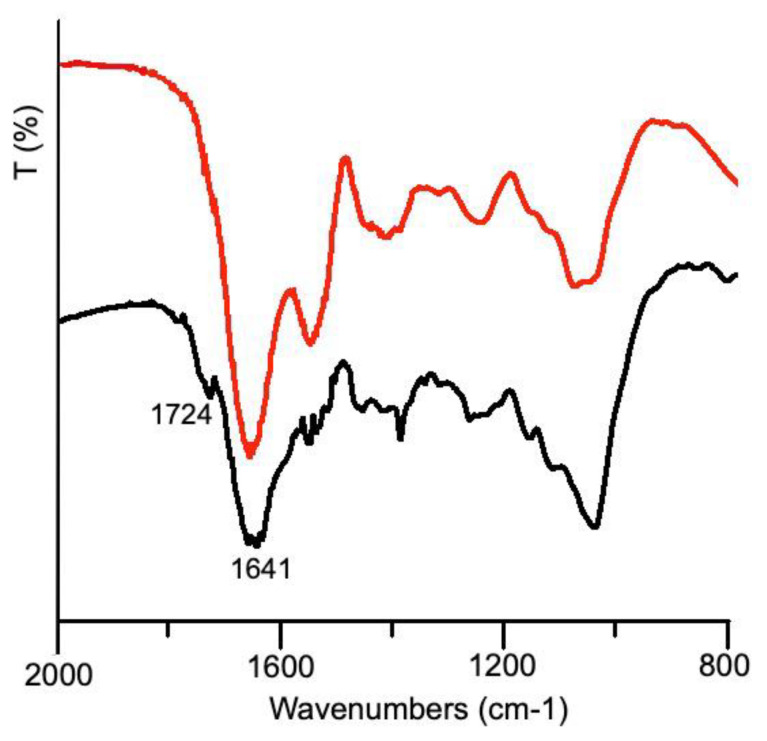
Infrared spectra of dialyzed snail slime (red line) and dried AgNPs collected from synthesis H.

**Figure 6 nanomaterials-12-03447-f006:**
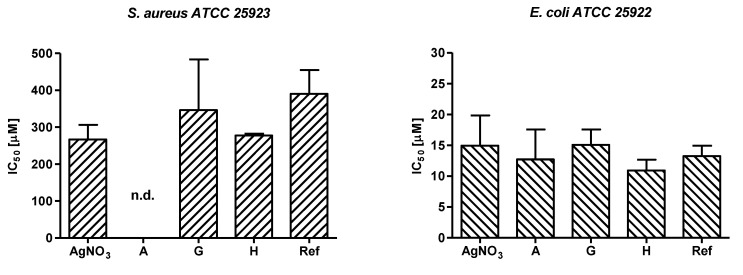
IC_50_ values (mean with standard deviation) obtained for the tested samples. The IC_50_ value for sample A was not determined (n.d.) on *S. aureus* as a consequence of the low starting concentration of the sample (300 µM) combined with the poor antibacterial activity of the Ag-based compounds on this Gram-positive strain. No statistical differences were observed when comparing the tested samples (see Appendix A).

**Table 1 nanomaterials-12-03447-t001:** Characterization of snail slime as received and after dialysis: evaluation of the dry residue and protein content. Each value is the mean of 5 tests and is reported with its standard deviation.

	Snail Slime as Received		Snail Slime after Dialysis	
Dry residue (%)	1.1 ± 0.1		0.04 ± 0.01	
Protein content	Liquid (mg/mL)	Solid freeze-dried (w %)	Liquid (mg/mL)	Solid freeze-dried (w %)
	1.80 ± 0.04	14 ± 3	0.36 ± 0.04	100 ± 2

**Table 2 nanomaterials-12-03447-t002:** Amount of reagents and temperature conditions used for each trial.

Sample	[AgNO_3_] (1.5 mL)	Snail Slime (mL)	Ultra-Pure Water (mL)	Temperature (°C)
A	10^−3^ M	1.0	2.5	25
B	10^−3^ M	1.0	2.5	60
C	10^−3^ M	2.0	1.5	25
D	10^−3^ M	2.0	1.5	60
E	10^−3^ M	0.5	3.0	25
F	10^−3^ M	0.5	3.0	60
G	10^−2^ M	1.0	2.5	25
H	10^−2^ M	2.0	1.5	25

## Data Availability

Not applicable here.

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
