# Peer review of "Effectiveness of Snail Slime in the Green Synthesis of Silver Nanoparticles"

_nanomaterials, 2022, doi:10.3390/nano12193447_

Round 1
Reviewer 1 Report
In the Manuscript “Effectiveness of snail slime in the green synthesis of silver nanoparticles” authors used Helix Aspersa’s snail slime to synthesize silver nanoparticles (AgNPs). Interestingly, it was possible to reduce silver ions solution to NPs by snail slime alone at room temperature and under neutral pH, which is the strong part of the work. Composition of the snail slime solution was thouroughly studied, role of its components in stabilizing AgNP as well as silver reduction mechanism were investigated. Finally, the obtained AgNPs were subjected to antibacterial properties study. This is a good and polished study with well-defined goal, sufficient experiment, and straight-forward discussion considering literature context.
I, however, am of the opinion, that the work requires powder XRD data for obtained AgNPs for this paper to be published. As so, minor revision. Please, provide X-ray diffraction data of the prepared AgNP powders to prove no silver oxide, halides or sulfide formation. Calculate crystalline size (coherent scattering region size) from XRD peaks form.
Can the snail slime be considered as standard reactive? Will it have different composition and properties if we use snail slime solutions from various manufacturers? Please, discuss.
Author Response
Please, see the attachment

Reviewer 2 Report
1. In abstract, the authors mentioned the large scale production; can this method fulfull this goal, what is the maximum production?
2. The size of nanoparticles was influenced by the synthesized conditions?
3. In antibacterial section, the antibacterial performance of Ag NPs and silver ions was comparable, why, it seems unreasonable.
4. Compared with other synthesized methods, the advantages of this method should be provided. For example, the references" J. Phys. Chem. Lett., 2022, 13, 29, 6721-6730; Small, 16(24):2000436 should be cited and discussed.
Reviewer 3 Report
Authors should add one paragraph under experimental methods like "Collection of Snail slime" which should include, the collection method, snail culture methods, temperature, ph, etc. It will help the researcher to further research on this topic.
Please add some closely related references here which are important. Like: DOI: 10.1109/TNB.2022.3186941;https://doi.org/10.1007/s00339-022-05784-7; DOI: https://doi.org/10.1166/jbn.2011.1344 and etc.
Round 2
Reviewer 2 Report
1. In figure 4, the scale bars is inconsistent, please revise it
2. Similari studies have been reported by Ravindra in Scientific reports, 2021, 11(1), 1-16. It is better to highlight the novelty of this studies in the manuscript.
